# Enhancement of Chest Radiograph in Emergency Intensive Care Unit by Means of Reverse Anisotropic Diffusion-Based Unsharp Masking Model

**DOI:** 10.3390/diagnostics9020045

**Published:** 2019-04-24

**Authors:** Sheng Chen, Yuantao Cai

**Affiliations:** Department of Automatic Control, School of Optical-Electrical and Computer Engineering, University of Shanghai for Science and Technology, 516 Jungong Road, Shanghai 200093, China; yuantao.cai@usst.edu.cn

**Keywords:** portable chest radiograph (CXR), intensive care unit (ICU), unsharp masking (USM), reverse anisotropic diffusion (RAD)

## Abstract

In intensive care units (ICUs), supporting devices play an important role, and the placement of these devices must be accurate, such as catheters and tubes. Taking portable chest radiograph (CXRs) for patients in ICU is a standard procedure. However, non-optimized exposure settings and misaligned body positions usually mean that portable CXRs are not in acceptable working condition. The purpose of this study was to enhance ICU CXRs to assist radiologists in the positioning of endotracheal, feeding, and nasogastric tubes in ICU patients. The unsharp masking model (USM) was a classical image enhancement technique. Because of the isotropic diffusion filter applied in this model, USM enhanced the edge information and noise simultaneously. In this paper, we proposed a reverse anisotropic diffusion (RAD)-based USM technique for enhancement of line structures in ICU CXRs. First, a RAD algorithm was applied to replace the Gaussian filter in the classical USM. The RAD algorithm only produced a smoothed image, in which edge information was smoothed while the noise was preserved. Then, the smoothed image was subtracted from the original image to produce the unsharp mask whereby only the edges were retained. Consequently, only edge information was enhanced in the final enhanced image by using the RAD-based USM model. The proposed method was tested for 87 ICU CXRs and the findings indicate that this approach can enhance image edges efficiently while suppressing noise.

## 1. Introduction

Chest X-ray radiographs (CXRs) are a standard imaging modality that accounts for about 40% of the total radiographs in a department of radiology. One-third of these CXRs are acquired by using the portable devices, such as computer radiograph (CR), digital radiograph (DR) in intensive care units (ICUs) [1]. Compared to the standard CXRs, patient respiratory and movement may lead to artifacts in the image because of the longer exposure time [2]. ICU CXRs usually present poor contrast and structure noise. Some false connections are exhibited between kinds of tubes, such as the endotracheal tube (ET), food tube (FT), and nasogastric tube (NT) in the image. These limitations of ICU CXRs make it complicated for interpretation [3]. The prediction of traumatic brain injury survival rates by data mining patient informatics processing software hybrid hadoop hive is proposed by Rodger [4].

Unsharp masking model (USM) is a classical image sharpening technique. It has widespread application and use, which include most of digital image processing software. The main idea of USM is simple. Firstly, an unsharp mask is created by subtracting a smoothed image from the corresponding original image. The unsharp mask only contains high-frequency information associated with edges and noise. Then a fraction of the mask is added to the original image to obtain an enhanced image. Gaussian filter is widely used to produce the unsharp mask. Nevertheless, it has two drawbacks: (a) The noise is emphasized in the regions of high details; (b) Fine details and important edges are sharpened equally because of the inherent properties of isotropic Gaussian filters [5]. A large number of researchers have attempted to address this problem. A modified version of unsharp masking was presented that was able to perform image smoothing, while not only preserving but also enhancing the salient details in images [6]. USM using quadratic filter for the enhancement of fingerprints in noisy background was discussed in [7]. An adaptive USM was presented in [8,9]. They discussed several popular techniques for improvement of USM, such as Laplacian filter, quadratic operator filter, higher order polynomial operator, and adaptive filter. Deng proposed a generalized USM algorithm using the exploratory date model as a unified framework [10].

In this paper, a reverse anisotropic diffusion (RAD)-based USM was proposed for enhancement of ICU CXRs. Firstly, the traditional anisotropic diffusion was introduced. Diffusion is a physical process, which involves the movement of a substance down a concentration gradient. Fick’s law can represent the diffusion as following
(1)y=−D×∇f
where ∇f is a concentration gradient, which causes a flux y, D is a symmetric matrix and represents the diffusion tensor. When y and ∇f are parallel, the diffusion is isotropic. If they are not parallel to each other, it will transfer to anisotropic diffusion. As known, diffusion abides the law of conservation of mass. It only transports matter instead of creating or destroying mass and can be described by the equation, represented by
(2)$$\partial_{t}f=div\left(y\right)$$
where t is time variable and div(y) is the divergence operator.

The following equation can be achieved by plugging Equation (1) into Equation (2), represented by
(3)$$\partial_{t}f=div\left(D\times \nabla y\right)$$

The diffusion equation was widely used for image processing in the past decades [11]. The linear diffusion process is represented by
(4)$$\frac{\partial I\left(x,y,t\right)}{\partial t}=div\left(\nabla I\right)\;,\;I\left(x,y,0\right)=f\left(x,y\right)$$
for which the solution is represented by
(5)$$I\left(x,y,t\right)=G_{\sqrt{2t}}\times f\left(x,y\right)$$
where $G_{\sqrt{2t}}\times f\left(x,y\right)$ represents the result of convolution by using a Gaussian filter with $\sigma =\sqrt{2t}$. The original image f(x,y) is set as the initial condition at time 0, (x,y) is the coordinate of an image pixel, t is the time variable, and ∇I is the gradient.

The result of smoothing by using a Gaussian filter with σ is identical to that by using a linear diffusion, which stops at time $t=0.5\delta^{2}$. In our proposed model, the linear diffusion was replaced by a non-linear diffusion procedure.

A famous anisotropic diffusion-Perona–Malik model was introduced [12]. They claimed to use the anisotropic diffusion instead of the classical linear diffusion, represented by
(6)$$\frac{\partial I\left(x,y,t\right)}{\partial t}=div\left(g\left(\|\nabla I\|\right)\nabla I\right)\;I\left(x,y,0\right)=f\left(x,y\right)$$
where g(‖∇I‖) is the diffusion coefficient function and it has two conditions: (1) g is a non-infinite function when ‖∇I‖ is zero; (2) g is a non-negative monotonically decreasing function.

In this model, fine details were smoothed and obvious edges were preserved due to the rapidly decreasing attributes of the diffusion coefficient function. Diffusions with different rates were studied in [13]. The model cannot achieve satisfied result when the image contains isolated noisy points because the noise introduces large oscillations of the gradient ‖∇I‖. The Perona–Malik model was improved by using a Gaussian smoothed diffusion coefficient function $g(\left\|\nabla I_{\sigma }\right\|)$ instead of the diffusivity g(‖∇I‖) [14]. Alvarez noted that Perona–Malik model did not have a clear geometric interpretation [14]. Therefore, he introduced a degeneration model, represented by
(7)$$\frac{\partial I\left(x,y,t\right)}{\partial t}=g\left(\|\nabla I\|\right)\|\nabla I\|div\left(\frac{\nabla I}{\left\|\nabla I\right\|}\right)$$
where $\|\nabla I\|div\left(\frac{\nabla I}{\left\|\nabla I\right\|}\right)$ is the degeneration term, and Equation (7) can be rewritten as
(8)$$\frac{\partial I\left(x,y,t\right)}{\partial t}=\frac{I_{xx}I_{y}^{2}-2I_{x}I_{y}I_{xy}+I_{x}^{2}I_{yy}}{I_{x}^{2}+I_{y}^{2}}$$

This model has a clear geometric meaning. It moves in the normal direction, which is proportional to its mean curvature.

The total variation (TV) minimization model was introduced in [15]. They proved that the mean curvature was the Euler–Lagrange equation derived from TV minimization.

The goal of classical anisotropic diffusion, such as Perona–Malik model or Degeneration model, is to remove noise and keep edges untouched. In USM, a RAD procedure is required in order to obtain the smoothed image, in which edges are smoothed while noise is preserved. This implies that classical anisotropic diffusion is unsuitable for our intended application. It needs to be modified for USM processing.

## 2. Materials and Methods

### 2.1. Databases of ICU CXRs

The Kodak portable CR 400 system was used in this study equipped at Xinhua Hospital, Shanghai, China. The ethic committee at Xinhua Hospital (Reference number 20130021, approval date: 21 May 2013) approves the database. All procedures performed in studies involving human participants were in accordance with the ethical standards of the institutional and national research committee and with the 1964 Helsinki declaration and its later amendments or comparable ethical standards. Informed consent was obtained from all individual participants included in the study. In Kodak portable CR system, the exposure index (EI) related to the logarithm of the incident exposure. In our experiment, the kVp, mAs and EI values were 80, 1.4, and 1770 respectively. Eighty-seven portable CXRs were collected from 20 patients using the Kodak CR 400 system. For every patient, two to nine images with a size of 2500 × 2048 pixels were obtained. The gray scale was 12 bits and the resolution was 0.171 mm × 0.171 mm per pixel.

We also used the Japanese Society of Radiological Technology (JSRT) dataset to test the proposed method. Every image was of 2048 × 2048 pixels and every pixel was of 0.175 mm × 0.171 mm in size. The gray scale of JSRT was also 12 bits.

### 2.2. USM

The USM model is a classical method for digital image processing. The edges can be enhanced through a procedure, which subtracts a diffused image from the corresponding original image. Let I(x,y), $I_{sm}\left(x,y\right)$, and $I_{en}\left(x,y\right)$ be the original image, the smoothed image, and the enhanced image, respectively. The unsharp masking model is represented by
(9)$$I_{en}\left(x,y\right)=I\left(x,y\right)+\alpha \times \left(I\left(x,y\right)-I_{sm}\left(x,y\right)\right)$$
where α is the weight of the un-sharp mask.

One of the disadvantages of USM model is that the gray scale value is sometimes out of the range. Since the generated unsharp mask may contain negative gray scale values and it causes a dark halo along the edges, which is undesirable. The other issue is that noise is enhanced in the final image in the regions near edges when an isotropic Gaussian filter is applied. To address these issues, we proposed an innovative method to achieve the unsharp mask, which reduced the side effects and noise amplification in classic USM.

### 2.3. Reverse Anisotropic Diffusion (RAD)

First, an energy functional was introduced and minimized.
(10)E(Idiffusion)=12‖Idiffusion−I‖2+α∫(1β+2)(‖∇Idiffusion‖2+μ)β+2dxdy
where E is the energy functional, μ is very small value to make sure that $\sqrt{{\left\|\nabla I_{diffusion}\right\|}^{2}+\mu }$ is not equal to zero, I is the original image; α>0, β=1.

This approach was deformed from TV minimization [16]. The first term in Equation (10) requires that the results approximate the original image.

Traditional anisotropic diffusion model, such as Perona–Malik model or Degeneration model attempted to remove as much noise as possible while keeping the edges intact. In this case, we developed a reverse diffusion-RAD algorithm, which only smoothed edges while keeping noise. When the smoothed image was subtracted from the original image, the edge reinforcement unsharp mask could be obtained. This mask was then overlapped with the original image to achieve the final enhancement image.

In the proposed model, edge needs to be defined explicitly. We use the same concept as in [12]. Classical anisotropic diffusion encourages piecewise smoothing in a region while preserving edges across the boundary, so the diffusion coefficient is set to be relative larger in the interior of region and smaller across the boundary. In our model, we reversely want to smooth the edges across the boundary rather than in the region. This approach requires that diffusion coefficient function g(‖∇I‖) is a monotonically increasing function. In addition, the diffusion parallel to the edge should be much weaker than the diffusion in an orthogonal direction to the edge.

In Figure 1, ξ and η represent the direction perpendicular and parallel to ∇u, respectively. For image, ξ is an edge with constant gray scale value and η is the flow-line with maximal gray scale value variation. Equation (6) can be rewritten as following
(11)$$\frac{\partial I}{\partial t}=g\left(\left\|\nabla I\right\|\right)I_{\xi \xi }+\phi^{\prime }\left(\left\|\nabla I\right\|\right)I_{\eta \eta }$$
where ϕ(‖∇I‖)=‖∇I‖g(‖∇I‖) is the influence function.

Equation (11) shows that if we want to have a stronger diffusion in the direction parallel to the gradient, g(‖∇I‖) must be monotonically increasing. On the other hand, the model should have a weaker diffusion in the direction perpendicular to the gradient.

In summary, there are two conditions for the RAD algorithm: (1) g(‖∇u‖) must be a monotonically increasing function; (2) Diffusion coefficient perpendicular to the edge should be larger than that parallel to the edge. Base on the two rules, we get $g(\left\|\nabla I\right\|)={\left\|\nabla I\right\|}^{\beta }$, where β is a positive integer (in our experiment, β was set to 1). We achieved the RAD algorithm, represented by
(12)$$\frac{\partial I\left(x,y,t\right)}{\partial t}=div({\left\|\nabla I\right\|}^{\beta }\times \nabla I$$

It can be interpreted as the gradient descent flow of the energy function, represented by
(13)E(‖∇I‖)=∫(1β+1)‖∇I‖β+2dxdy
‖∇I‖ could be zero in flat areas and a small constant should be used to avoid this problem (in our experiment, it is set to be 0.001). Eventually, the objective is to minimize Equation (10) and the corresponding Euler–Lagrange equation is as following
(14)$$T(I_{diffusion})=I_{diffusion}-I+\alpha \times L\left(I_{diffusion}\right)\times I_{diffusion}=0\;\left(x,y\right)\in \Omega$$
where Ω is the image domain, and L is an operator such that $L\left(I_{diffusion}\right)=-div\left({\left\|\nabla I_{diffusion}\right\|}^{\beta }\times \nabla \omega \right)$.

Equation (14) can be expressed as
(15)$$\left[D+\alpha L\left({I_{diffusion}}^{m}\right)\right]{I_{diffusion}}^{m+1}=I\;m=0,1,2,\ldots$$
where *D* represents an identity matrix.

The half-pixel and central difference discrete methods are utilized to discretize Equation (15).
(16)$$I_{x,y}^{m+1}+\alpha \left[sI_{x,y}^{m+1}-c_{1}I_{x-1,y}^{m+1}-c_{2}I_{x+1,y}^{m+1}-c_{3}I_{x,y-1}^{m+1}-c_{4}I_{x,y+1}^{m+1}\right]=I_{x,y}$$
where
$$c_{1}=\sqrt{{\left(I_{x,y}^{m}-I_{x-1,y}^{m}\right)}^{2}+\frac{1}{4}{\left(I_{x-1,y+1}^{m}-I_{x-1,y-1}^{m}\right)}^{2}+\beta }$$
$$c_{2}=\sqrt{{\left(I_{x+1,y}^{m}-I_{x,y}^{m}\right)}^{2}+\frac{1}{4}{\left(I_{x,y+1}^{m}-I_{x,y-1}^{m}\right)}^{2}+\beta }$$
$$c_{3}=\sqrt{{\left(I_{x,y}^{m}-I_{x,y-1}^{m}\right)}^{2}+\frac{1}{4}{\left(I_{x+1,y-1}^{m}-I_{x-1,y-1}^{m}\right)}^{2}+\beta }$$
$$c_{4}=\sqrt{{\left(I_{x,y+1}^{m}-I_{x,y}^{m}\right)}^{2}+\frac{1}{4}{\left(I_{x+1,y}^{m}-I_{x-1,y}^{m}\right)}^{2}+\beta }$$
$$s=c_{1}+c_{2}+c_{3}+c_{4}$$

There are many algorithms to solve the linear equations, such as Gaussian-Jordan elimination, LU (lower-upper) decomposition [17]. They are efficiency for linear equation analysis. However, these techniques can hardly be used for solving Equation (16) because the size of the images in our portable CXRs database is 2500×2048, and the images are too large to be handled by a direct implementation. We tried to store the results by using some special storage schemes and some methods can be found in [18]. In this paper, the indexed storage scheme was chosen [19].

The conjugate gradient (CG) is a traditional algorithm for solving positive definite linear system. The minimization of the quadratic function can be represented by
(17)$$f\left(x\right)=\frac{1}{2}x\times A\times x-b\times x$$

The gradient is
(18)g(x)=A×x−b

Many algorithms based on the CG method [20,21] have been researched and developed. The primary shortcomings of these algorithms are that the matrix not only be symmetric, but also be positive definite. It means that the CG method is not suitable. Then, the bi-CG algorithm was developed to solve this problem. The convergence rate of the CG method relies on the distribution of the eigenvalues of matrix A in Equation (18). In order to get better eigenvalues, the condition number should be small. Furthermore, a small condition number often leads to a rapid convergence [22]. There was a technique called preconditioning can be applied for Equation (18) to produce a better condition number. The original CG method has a good performance if the matrix A is ‘close’ to the identity matrix. Therefore, the following preconditioning matrix is applied for solving the equation.
(19)$$({\tilde{A}}^{-1}\times A)\times x={\tilde{A}}^{-1}\times b$$
where ${\tilde{A}}^{-1}\times A\approx 1$, b is a vector and matrix $\tilde{A}$ is the preconditioner. Here, we use the diagonal part of the matrix A as the preconditioner. For demonstration of the preconditioned bi-conjugate gradient algorithm (PBCG), we introduce several vectors, $\bar{r_{k}}$, $r_{k}$, $p_{k}$, $z_{k}$, and $\bar{z_{k}}$, k=1,2,3…. We also give an initial solution for x. $z_{k}$ and $\bar{z_{k}}$ are defined by ${\tilde{A}}^{-1}\times z_{k}=r_{k}$ and ${\tilde{A}}^{-1}\times \bar{z_{k}}=\bar{r_{k}}$. Then, the following iteration can be represented by
$$\alpha_{k}=\frac{\bar{r_{k}}\times z_{k}}{\bar{p_{k}}\times A\times p_{k}}$$
$$\beta_{k}=\frac{\bar{r_{k+1}}\times z_{k+1}}{\bar{r_{k}}\times z_{k}}$$
$$r_{k+1}=r_{k}-\alpha_{k}\times A\times p_{k}$$
$$\bar{r_{k+1}}=\bar{r_{k}}-\alpha_{k}\times A^{T}\times \bar{p_{k}}$$
(20)$$p_{k+1}=z_{k}+\beta_{k}\times p_{k}\times \bar{p_{k+1}}$$
where $r_{0}=b-A\times x_{0},\;p_{0}=z_{0}-{\tilde{A}}^{-1}\times r_{0}$. The sequence of estimated solution is $x_{k+1}=x_{k}+\alpha_{k}\times \beta_{k}$. The smoothed image can be achieved by using PBCG for solving the anisotropic diffusion equation.

### 2.4. Improved USM

Once the edge-smoothed image is acquired by using RAD algorithm (Figure 2h), we subtract it from the original image to get the unsharp mask (Figure 2i), and then it is added to the original image (Figure 2j). Figure 2h shows that the edges in the image were smoothed and the noise was kept when the RAD algorithm was used, while the noise in Figure 2b were smoothed as well as the edges. In the final enhanced image, only the edges were enhanced in Figure 2j, while noise in Figure 2d, where the Gaussian smoothing method was used, was more obvious than that in the original image. Figure 2e shows the smoothed image acquired by using classical anisotropic diffusion. Only noise was enhanced in Figure 2g.

## 3. Results

Before we apply our proposed method to the ICU CXRs database, we need to establish an objective measurement method to evaluate the improvement. A method called the measurement of enhancement by entropy evaluation (EMEE) was presented in [23]. It is commonly used for image processing quality evaluation. We used this method to evaluate our RAD-based USM method. This evaluation method split the image I(i,j) into k1×k2 blocks $w_{k,l}(i,j)$ of size $l_{1}\times l_{2}$ and find the local maximum $I_{\max;k,l}$ and minimum $I_{\min;k,l}$ within each block. EMEE can be calculated as
(21)$$EMEE=\frac{1}{k1\cdot k2}\sum_{l=1}^{k2}\sum_{k=1}^{k1}\frac{I_{\max;k,l}^{w}}{I_{\min;k,l}^{w}+c}\log\frac{I_{\max;k,l}^{w}}{I_{\min;k,l}^{w}+c}$$

In our experiment, pixels with zero intensity were frequently present. Therefore, a small positive constant was selected to keep the denominator from having a value of zero (c is set to 30 in our experiment). In addition, we set $l_{1}\times l_{2}$ to be 40 × 40, the larger value of EMEE, the better result. Nevertheless, we must be aware that it will affect the local maximum and local minimum, if there is too much noise, thus affecting the result of EMEE.

In order to compare our method with traditional USM algorithm, we applied both the traditional USM method and our approach for the portable CXRs Database.

For traditional USM method, we have chosen different kernel size for Gaussian filter (15×15, 19×19, and 23×23 separately). Figure 3 shows a CXR from the portable CXRs database. The contrast of line structure in the spine and heart area is limited, and it is hard to find the tips of ET and NT. Enhancement is necessary for ICU CXR. Figure 4 shows the enhanced results obtained by using traditional USM. The EMEE value for the original image was 0.491. After enhancement, the EMEE values were improved to 0.841 (Figure 4c), 0.980 (Figure 4f), and 1.106 (Figure 4i), corresponding to the Gaussian kernel size of 15×15, 19×19, and 23×23, respectively. Although, the EMEE values were improved, some important edge information still did not show clearly in the image, especially when the kernel size was smaller. When the kernel size of Gaussian increased, the noise and the edges were enhanced simultaneously. As a result, Figure 4c,f,i did not have high contrast, which was predictable based on the properties of the traditional USM algorithm.

In the RAD-based USM method, the parameter α dramatically influences the result. Several different values for this parameter were set so that we can analyze the effect of this term according to various parameters. Firstly, we started with a very small value of 0.001. As we can see in Figure 5a–c (EMEE = 0.587), the result was not desirable since there was the presence of noise. In addition, it did not show a clear contour. As we magnified the parameter α, the result was improved. As α approached approximately 0.02, the EMEE value of the result was 1.24 (Figure 5d–f), which was greater than that for the traditional method. The noise was less than that shown in Figure 4 and the contour was clearer. When the parameter α became bigger, for example, 0.2 (Figure 5g–i), the EMEE value was 1.43, which was greater than the EMEE value for α equal to 0.02. More structural information was enhanced in the unsharp mask (Figure 5h). Indeed, we have observed that the contours became thicker. The edge of the ribs became wider which was a good outcome. It was also noticed that less noise was presented in the unsharp mask. The reason for this phonomenon is that the diffusion is too strong. We set α with 0.02 and applied our RAD-based USM method for each CXR in the portable CXRs database. Then, we calculated EMEE values for every CXR. To facilitate a comparison, we also calculated the EMEE values for every CXR by using the traditional USM method for the portable CXRs database. Figure 6 shows that the RAD-based USM method usually has a larger EMEE value than that the Gaussian filter-based USM method has. Objective and subjective evaluation methods show that the proposed method is acceptable.

We also conducted the SNR for the portable CXR database for evaluation of the improvement. The average SNR value for all the original portable CXRs was 105.1 and it was improved to 311.4 after enhancement by use of the RAD-based USM method. Table 1 shows the docotors’ evaluation results. The test values tell us that the enhancement method substantially improved the 5-point score for all three clinicians comparing to original images.

We also applied our method for the JSRT database in a similar manner for that of the portble CXRs database. Figure 7 shows an original image from JSRT database. We can see from Figure 8a that much more noise was created in the unsharp mask by using the Gaussian kernel. The EMEE value for the original CXR was 1.453. Although the EMEE value was improved to 5.643 for the final enhancement image, the noise in Figure 8a was also added in the final enhancement.

When, we applied our RAD-based USM method to this image from the JSRT database, the diffusion parameter *α* was set to 0.02. Most of the important edge information was reinforced in the unsharp mask with less noise (as shown in Figure 8b).

## 4. Discussion

We presented an image enhancement method for CXRs in ICU by using RAD-based USM. It is acknowledged that several factors must be considered when the comparisons are made. Based on previous results, we conclude that our RAD-based USM method improves the enhancement of line structures such as endotreacheal tube (ET), feeding tube (FT), and nasogastic tube (NT) in CXRs. We also notice that it is difficult to get the optimized parameter in our method, because portable CXRs use different dose setting, positioning etc.

The interpretation of CXRs is usually challenging, especially for the inexperienced radiologists. Therefore, we developed an improved RAD-based USM to enhance the image and improved their diagnostic quality. By using our proposed method, radiologists can improve their confidence and ability to characterize lung parenchyma and subtle pathological changes.

In this paper, we obtained different EMEE values by using different diffusion parameters in the RAD algorithm. The parameters were fixed for diffusion, which corresponding to the highest performance for image enhancement for the portable CXRs and JSRT databases, respectively. However, these parameters may not be optimized for other CXR database. There is a potential improvement for the RAD algorithm, because the contrast of CXR in each database is different for case to case with the exposure dose. Thus, the diffusion parameter of RAD should be adaptive optimized based on each case.

Although, the evaluation standard of EMEE is more consistent with subjective visual effect, it is also the most commonly used image processing evaluation standard. However, some other evaluation methods may be applied in our future work. For example, the question of whether the performance of computer aided detection system for nodule detection can be improved by the use of enhancement CXR and the extent of the performance improvement needs to be addressed.

The time for processing one image with our enhancement method was 10.1 s on a PC-based workstation (Intel Pentium 2.4 GHz processor with a 3 GB memory).

## 5. Conclusions

We developed a reverse anisotropic diffusion-based unsharp masking model. A reverse anisotropic diffusion model was applied to replace the Gaussian filter in the traditional USM. The reverse anisotropic model only smooth edge information while keeping noise in the image. The smoothed image was then subtracted from original image to produce an unsharp mask where only edges were retained. Consequently, only edge information was enhanced in the sharped image. The findings indicated that this method enhanced image edges efficiently, simultaneously preserving noise.

## Figures and Tables

**Figure 1 diagnostics-09-00045-f001:**
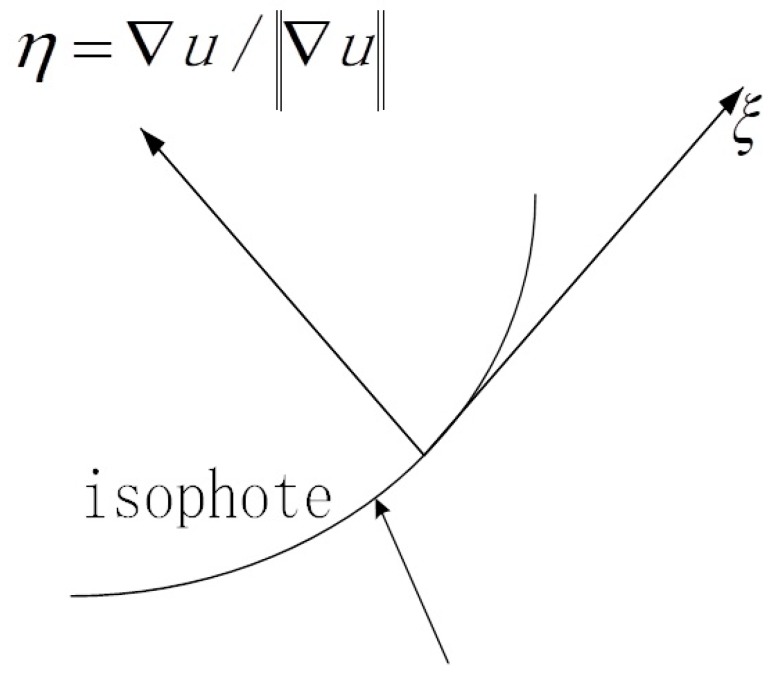
ξ and η denote the direction of perpendicular and parallel to ∇u, respectively.

**Figure 2 diagnostics-09-00045-f002:**
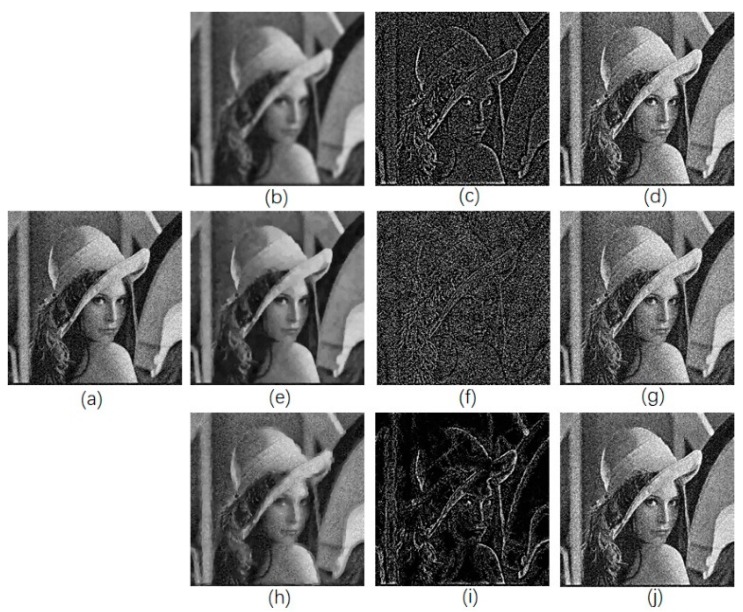
Image enhancement by use of different model. (**a**) Original image; (**b**) image smoothed by Gaussian kernel; (**c**) unsharp mask by subtracting (**b**) from (**a**); (**d**) enhanced image by adding (**c**) with (**a**); (**e**) smoothed image by use of traditional anisotropic diffusion; (**f**) Unsharp mask; (**g**) enhanced image; (**h**) smoothed image by use of reversed anisotropic diffusion; (**i**) Unsharp mask; (**j**) enhanced image.

**Figure 3 diagnostics-09-00045-f003:**
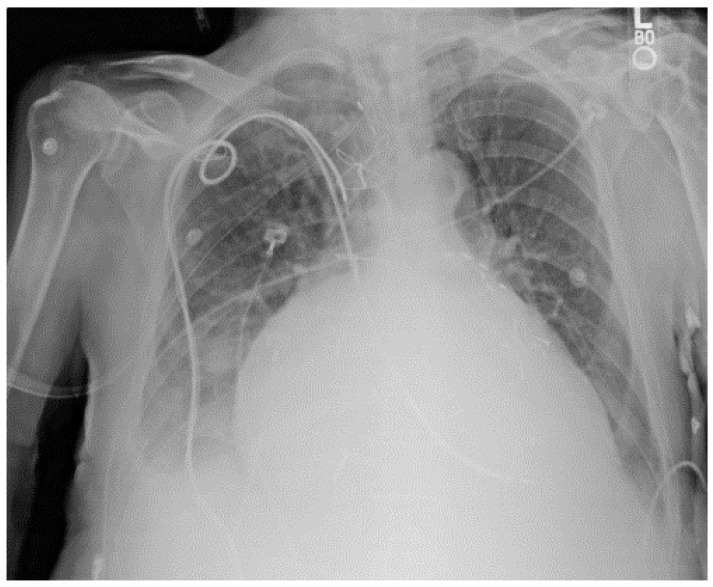
An image from portable chest radiograph (CXRs) database.

**Figure 4 diagnostics-09-00045-f004:**
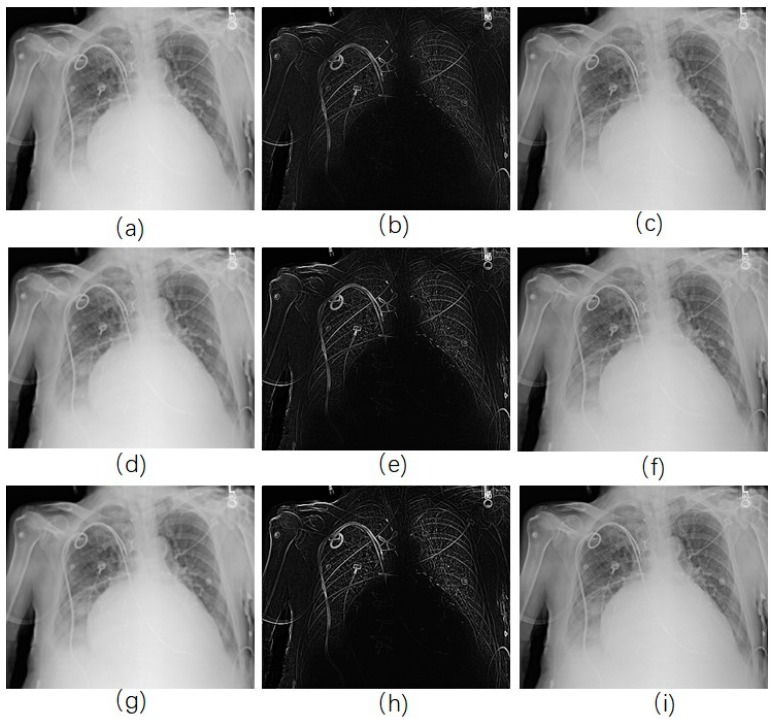
CXR enhancement by use of traditional USM model. (**a**,**d**,**g**) were the smoothed images by use of 15×15, 19×19, 23×23 Gaussian kernel, respectively; (**b**,**e**,**h**) were the corresponding Unsharp masks to (**a**,**d**,**g**); (**c**,**f**,**i**) were the corresponding enhanced images.

**Figure 5 diagnostics-09-00045-f005:**
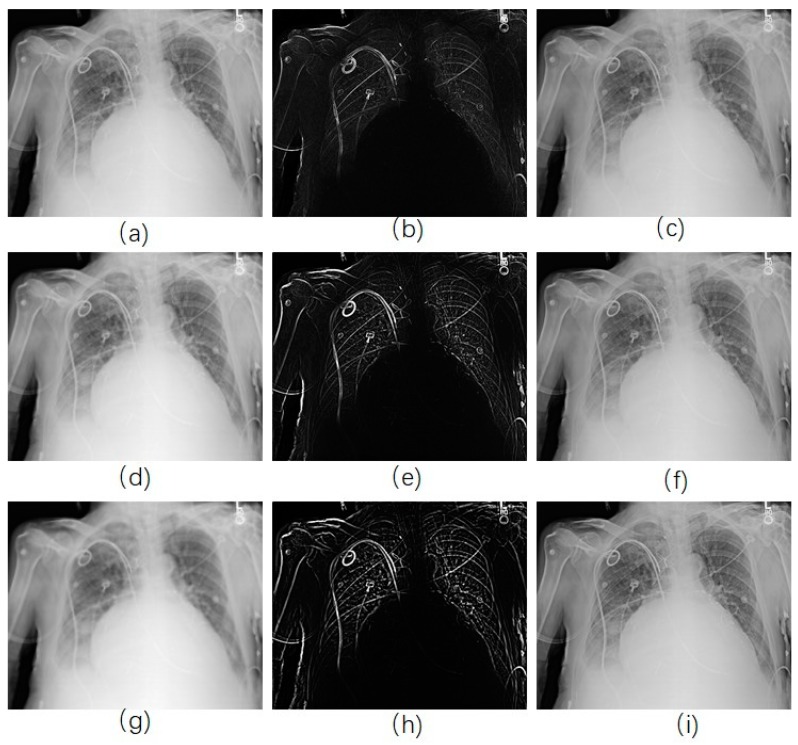
CXR enhancement by use of reverse anisotropic diffusion (RAD) -based USM. (**a**,**d**,**g**) were the smoothed images with diffusion parameter 0.001, 0.02, 0.2, respectively; (**b**,**e**,**h**) were the corresponding unsharp masks to (**a**,**d**,**g**); (**c**,**f**,**i**) were the corresponding enhanced images.

**Figure 6 diagnostics-09-00045-f006:**
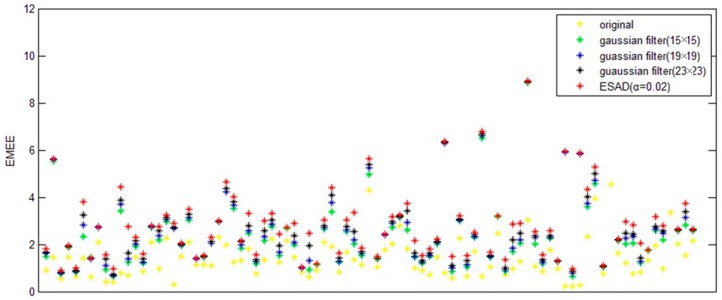
the EMEE value for the original images, results of our method and USM model, respectively, the kernel sizes of Gaussian filter are 15 × 15, 19 × 19, 23 × 23.

**Figure 7 diagnostics-09-00045-f007:**
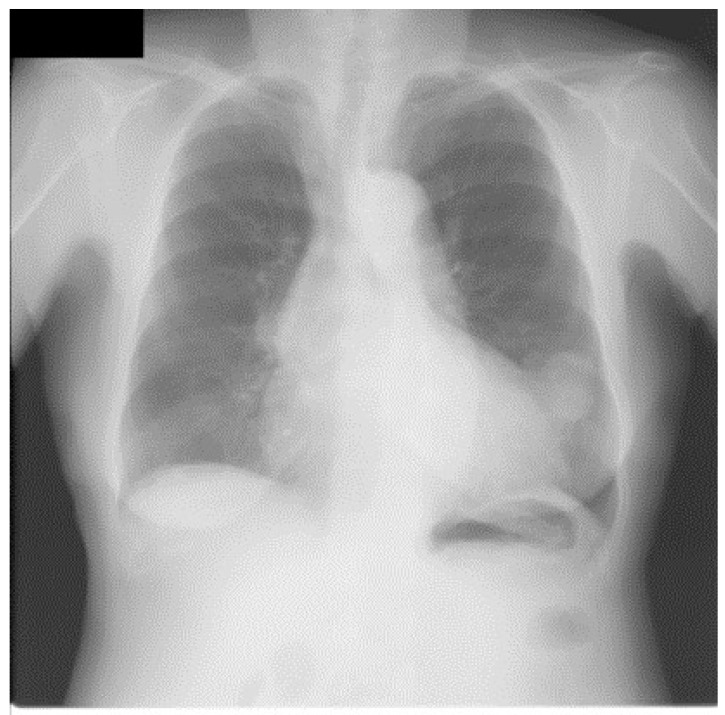
An image from Japanese Society of Radiological Technology (JSRT) database.

**Figure 8 diagnostics-09-00045-f008:**
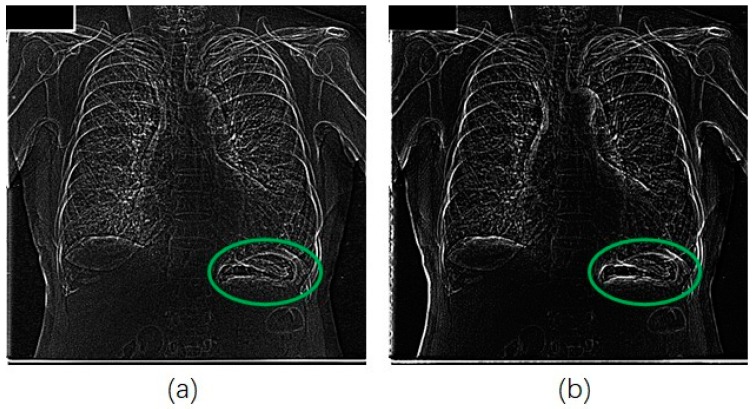
Unsharp masks for image of Figure 7. Green circles are the stomach area, (**a**) mask from Gaussian difussion; (**b**) mask from RAD.

**Table 1 diagnostics-09-00045-t001:** The evaluation result.

Image	Radiologist	Score					Average
		5	4	3	2	1	
Enhanced portable CXRs	Clinician 1	55	21	11	0	0	4.51
	Clinician 2	50	23	13	1	0	4.40
	Clinician 3	57	21	10	0	0	4.58
Original CXRs	Clinician 1	30	31	13	13	0	3.89
	Clinician 2	24	34	12	12	5	3.69
	Clinician 3	32	33	10	12	0	3.97

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
