# Peer review of "Enhancement of Chest Radiograph in Emergency Intensive Care Unit by Means of Reverse Anisotropic Diffusion-Based Unsharp Masking Model"

_diagnostics, 2019, doi:10.3390/diagnostics9020045_

Round 1
Reviewer 1 Report
The method proposed in the paper gives a contribtion on the enhancement of the X-ray images,
The noise seems to be reduced.
However the method parameters must be choosen carefully.
Some comments on the presentation of the work:
1) the english must be improved in some of the sentences.
2) the authors must guide the reader especially in the mathematical path; I invite the authors to go through the equations and detail each parameter in te text.
Some comments on the work:
1) About the subjectivity of image inerpretation: is it possible having a statistic of redeability of the enhanced images by doctors interview?
2) From the results it is clear that the EMEE value is increased more or less in each processed image; Is it possible having a comparison by other objective parameters (e.g. MTF, SNR, FFT, etc...)?
Author Response
Responses to Reviewers
We would like to thank you for reviewing our manuscript. As suggested, we have improved our manuscript by following your comments. Please refer to our detailed responses below which address all of your comments.
The method proposed in the paper gives a contribtion on the enhancement of the X-ray images,
The noise seems to be reduced.
Our response: Thanks for your comments.
However the method parameters must be choosen carefully.
Our response: We agree with you that there are some parameters that should be choosen carefully. And this is our future work how to optimize these parameters.
1) the english must be improved in some of the sentences.
Our response: We have improved the english writing based on your suggestion.
2) the authors must guide the reader especially in the mathematical path; I invite the authors to go through the equations and detail each parameter in the text.
Our response: Thanks for your invitation. We have gone through the equations and details in the text.
1) About the subjectivity of image inerpretation: is it possible having a statistic of redeability of the enhanced images by doctors interview?
Our response: Yes, there is a statistic of redeabliity of the enhanced images by doctors interview.
2) From the results it is clear that the EMEE value is increased more or less in each processed image; Is it possible having a comparison by other objective parameters (e.g. MTF, SNR, FFT, etc...)?
Our response: Yes, we have conducted the SNR parameter for evaluation of the processed image.
Reviewer 2 Report
The following is my review of the manuscript, "Enhancement of Chest Radiograph in Emergency Intensive Care Unit by Means of Reverse Anisotropic Diffusion Based Unsharp Masking Model.” - 47616. The technical components of the paper are sound and the manuscript “proposed a reverse anisotropic diffusion (RAD) based USM technique for enhancement of line structures in ICU CXRs.” The paper makes a valuable contribution to the literature and the experiments are of value to the readers of "Diagnostics". The paper's aims and scope match those of Diagnostics; therefore, the topic is appropriate for this journal. The proposal is appealing and interesting, and the method deserves consideration. Further, I would suggest that the author investigate and incorporate the following 2016 reference that would assist the author in contrasting, investigating and incorporating a similar optimization of production processes model: Rodger, J. A. (2016). Discovery of Medical Big Data Analytics: Improving the Prediction of Traumatic Brain Injury Survival Rates by Data Mining Patient Informatics Processing Software Hybrid Hadoop Hive. Informatics in Medicine Unlocked., 1C (17-26)., doi: 10.1016/j.imu.2016.01.002. The previously mentioned strengths of the manuscript suggest that the paper can be considered to be of high quality and also be of sufficient value to the readership of "Diagnostics", once the paper is thoroughly edited and the mentioned literature is investigated and incorporated into the manuscript. Therefore, based on the foregoing discussion, I would recommend that this paper be accepted, after reviewing and integrating the concepts and the citation that were referred to above.Author Response
Responses to Reviewers
We would like to thank you for reviewing our manuscript. As suggested, we have improved our manuscript by following your comments. Please refer to our detailed responses below which address all of your comments.
I would suggest that the author investigate and incorporate the following 2016 reference that would assist the author in contrasting, investigating and incorporating a similar optimization of production processes model:
Rodger, J. A. (2016). Discovery of Medical Big Data Analytics: Improving the Prediction of Traumatic Brain Injury Survival Rates by Data Mining Patient Informatics Processing Software Hybrid Hadoop Hive. Informatics in Medicine Unlocked., 1C (17-26)., doi: 10.1016/j.imu.2016.01.002.
Our response: Thanks for your comments. We have investigated and cited this paper in our manuscript.